# The Impact of Permafrost Degradation on Lake Changes in the Endorheic Basin on the Qinghai–Tibet Plateau

**Wenhui Liu [1], Changwei Xie [2,*], Wu Wang [2], Guiqian Yang [2], Yuxin Zhang [2], Tonghua Wu [2], Guangyue Liu [2], Qiangqiang Pang [2], Defu Zou [2] and Hairui Liu [3]**

1   Department of Geological Engineering, Qinghai University, Xining 810016, China; liuwenhui222@lzb.ac.cn
2   Cryosphere Research Station on the Qinghai-Tibet Plateau, State Key Laboratory of Cryosphere Sciences, Northwest Institute of Eco-Environment and Resources, Chinese Academy of Sciences, Lanzhou 730000, China; wangwu@lzb.ac.cn (W.W.); guiqianyang@163.com (G.Y.); zhangyuxin201108@163.com (Y.Z.); thuawu@lzb.ac.cn (T.W.); liuguangyue@lzb.ac.cn (G.L.); qqpang@lzb.ac.cn (Q.P.); zoudf09@lzu.edu.cn (D.Z.)
3   College of Eco-Environmental Engineering, Qinghai University, Xining 810016, China; lhrbotany@163.com
*   Correspondence: xiecw@lzb.ac.cn

**Abstract:** Lakes on the Qinghai–Tibetan Plateau (QTP) have experienced significant changes, especially the prevailing lake expansion since 2000 in the endorheic basin. The influence of permafrost thawing on lake expansion is significant but rarely considered in previous studies. In this study, based on Landsat images and permafrost field data, the spatial-temporal area changes of lakes of more than 5 km$^2$ in the endorheic basin on the QTP during 2000–2017 is examined and the impact of permafrost degradation on lake expansion is discussed. The main results are that permafrost characteristics and its degradation trend have close relationships with lake changes. Lake expansion in the endorheic basin showed a southwest–northeast transition from shrinking to stable to rapidly expanding, which corresponded well with the permafrost distribution from island-discontinuous to seasonally frozen ground to continuous permafrost. A dramatic lake expansion in continuous permafrost showed significant spatial differences; lakes expanded significantly in northern and eastern continuous permafrost with a higher ground ice content but slightly in southern continuous permafrost with a lower ground ice content. This spatial pattern was mainly attributed to the melting of ground ice in shallow permafrost associated with accelerating permafrost degradation. Whereas, some lakes in the southern zones of island-discontinuous permafrost were shrinking, which was mainly because the extended taliks arising from the intensified permafrost degradation have facilitated surface water and suprapermafrost groundwater discharge to subpermafrost groundwater and thereby drained the lakes. Based on observation and simulated data, the melting of ground ice at shallow depths below the permafrost table accounted for 21.2% of the increase in lake volume from 2000 to 2016.

**Keywords:** lake expansion; permafrost degradation; ground ice melting; endorheic basin; Qinghai–Tibetan Plateau

## 1. Introduction

The Qinghai–Tibetan Plateau (QTP) is the origin of many rivers and lakes and was the source of some large rivers in China and east Asian, and is therefore known as the "Asian water tower". The QTP has the largest lake numbers and areas in China; it contains more than 1000 lakes larger than 1 km$^2$ with a total area of ~41,800 km$^2$, accounting for ~39% and ~51% of the total lake numbers and areas in China, respectively [1]. In recent decades, the QTP has experienced significant climate changes, mainly characterized by a warming–wetting climate [2], accelerating glacier melting [3]

and permafrost thawing [4], which resulted in substantial lake changes in area [5–8], as well as changes in water levels [9,10] and lake volumes [11–13], especially the prevailing lake expansion since 2000 in the endorheic basin on the QTP [14,15]. Many previous studies focused on lake expansion and its influencing factors, but the main driver behind rapid lake expansion was still under debate. A consensus has formed that the increase in precipitation played a dominant role in the expansion of most QTP lakes [11–13,16], while some scholars have suggested that glacier meltwater has contributed significantly to lake growth [17,18]. As one of the significant components of the cryosphere system, permafrost plays a more and more significant role in influencing the hydrological regime and water resource in cold regions. Permafrost is widely distributed over the QTP with a total area of $1.06 \times 10^6$ km$^2$, accounting for 40.2% of the total area of the QTP [19], and the ground ice content in permafrost is 9528 km$^3$ [20]. The statistics show that the lake numbers and areas in the permafrost account for 68.8% and 40.8% of the total lake numbers and areas on the QTP. However, the effect of permafrost thawing on lake change has been rarely considered in previous studies.

Permafrost degradation due to rising air temperature on the QTP has been confirmed by many researchers: the decreasing permafrost area [19], rising permafrost temperature [21,22], increasing active layer thickness (ALT) [23–25], increasing lower limit of permafrost [26] and decreasing depth of the zero annual amplitude of the ground temperature of the permafrost [27,28]. Under the circumstances of permafrost degradation, the melting of ground ice and releasing of soil water content in permafrost were expected to provide more water resources to impact groundwater dynamics, enhance the hydraulic exchange between the surface water and groundwater and participate in the regional water cycle. In addition, the degradation of permafrost, including increasing ALT and reducing permafrost thickness, will lead to increases in aquifer thicknesses and surface water infiltration amounts, which changed the recharge, discharge and movement patterns of the surface water and groundwater, and further resulted in enlarging the groundwater storage capacity, increasing winter discharge [29–33], slowing down the recession flow process and decreasing the ratio of maximum to minimum discharge ($Q_{max}/Q_{min}$) in permafrost catchments [34–38]. Recent researches quantitatively indicated that meltwater from permafrost degradation were responsible for 20% of the outlet river water during the flood season in the Shiyang River [39], for 21% of the outlet runoff in the Hulugou River basin at the central Qilian Mountains [40] and for 24% ± 2.4% of the mean runoff in the Gulang River basin [41]. This hydrological effect of permafrost degradation was detected commonly in 39 QTP subbasins [34], eight inland river catchments in the Qilian Mountains [35] and three typical basins in the northern region of China [37]. All those researches suggested that permafrost degradation has a profound effect on the hydrological process in permafrost basins, giving rise to a significant influence on river runoff and lake changes.

However, at present, because of a lack of permafrost field data and the great difficulty in monitoring permafrost degradation–melt water release process on the QTP due to the hard environment, the impact of permafrost degradation on lake changes, in particular the contribution of permafrost degradation to lake expansion, has been seldomly studied, which has limited our understanding of the permafrost-hydrology process on the QTP. Therefore, in this study, based on Landsat images and permafrost field data, we firstly examined the spatial-temporal change in lake area in the endorheic basin on the QTP from 2000 to 2017, then discuss the spatial relationship between the permafrost characteristics and lake change and finally calculate the contribution of permafrost degradation to lake expansion.

## 2. Data and Method

### 2.1. Study Area

The endorheic basin is located in the northwestern QTP, covering the region in the south of the Kunlun Mountains, the west of the Hoh Xil Mountains and Tanggula Mountains and the north of the Gangdise Mountains and Nyainqentanglha Mountains, which is approximately consistent with the Qiangtang Plateau. The endorheic basin has a total area of $0.7 \times 10^6$ km$^2$ (accounting for 26.7%

of the entire QTP), with a width of 950 km from north to south and a length of 1359 km from east to west. It consists of 400 endorheic drainage basins with an average elevation of about 4894 m a.s.l. (Figure 1). The climate across the endorheic basin ranges from temperate arid to subarctic semi-humid. Dominated by the Indian monsoon in summer and cold-dry westerlies in winter, the climate indicates a strong seasonal variation [7]. The mean annual air temperature is near or below 0 °C. The mean annual precipitation is 50~300 mm; at least 60%~90% of the total annual precipitation falls between June and September, while less than 10% falls in winter between November and February. It is windy in winter and spring; approximately 200 days with a wind speed of more than 17 m/s has occurred. The endorheic basin is characterized by lack of surface runoff, a sparse river network and more seasonal rivers. The endorheic basin is a typical alpine frost sparse steppe, accounting for 64.5% of the total area. Most of the lakes on the QTP are located in the endorheic basin; it contains approximately 780 lakes larger than 1 km$^2$, which accounted for more than 73% of the total lake numbers and areas [7]. Several large lakes more than 1000 km$^2$, such as Seling Co (2405 km$^2$), Ayakekumu Lake (1041 km$^2$), Chibuzhang Co (1073 km$^2$) and Nam Co (2022 km$^2$), were located in this basin.

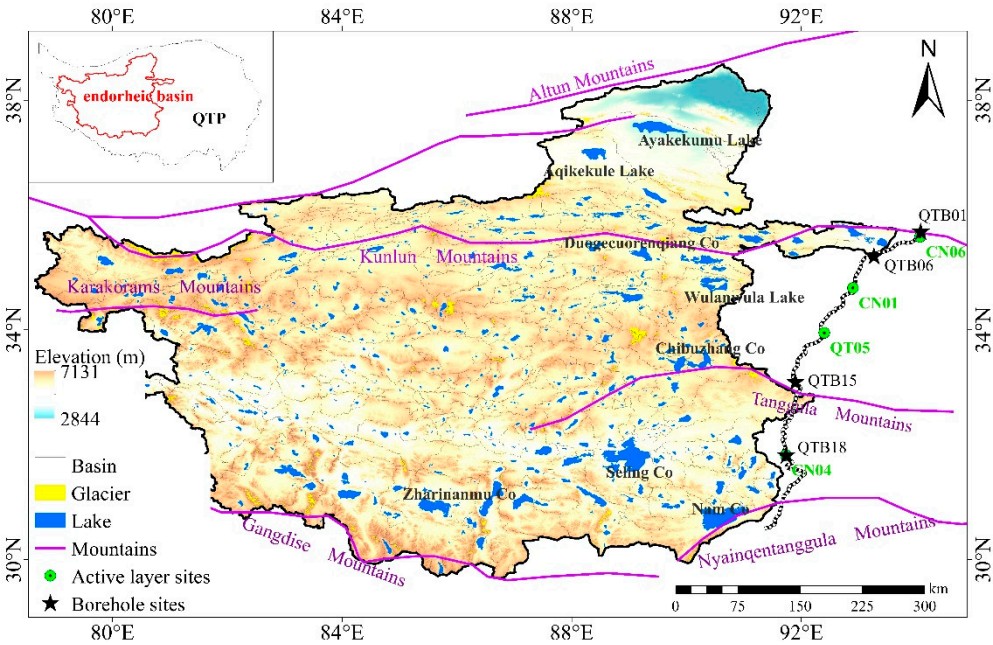

**Figure 1.** Schematic representation of the study area. The glacier data was from the Second Glacier Inventory Dataset of China (http://westdc.westgis.ac.cn). Lake sizes was from 2017 in this paper. The Digital Elevation Model was loaded from ASTER GDEM version 2.0 with a spatial resolution of 30 m (http://www.gscloud.cn), and the drainage basins were also derived from ASTER GDEM version 2.0.

*2.2. Data and Processing*

2.2.1. Satellite Images for Extracting Lake Area

Landsat TM (Thematic Mapper), ETM+ (Enhanced Thematic Mapper Plus) and OLI (Operational Land Imager) images (http://glovis.usgs.gov) with small fractions of cloud coverage (<10%) were chosen to extract the lake areas for a time series of areal changes between 2000 and 2017. All of these images with a spatial resolution of 30 m were acquired in October because the lake area in this month was relatively stable. The data window was also extended to September or November when images were not available in October. A total of 684 scenes covering the study area (38 scenes for each year) were used to delineate the lakes in the endorheic basin from 2000 to 2017. Lake boundaries were extracted in the false color compositions (bands 5, 4, and 3 as red, green, and blue for Landsat TM/ETM+, respectively, and bands 7, 6 and 5 for Landsat OLI) of the raw Landsat images for each lake using the ENVI 5.3 software. Then, visual examination and manual editing of lake boundaries

were conducted to delineate the lakes in ArcGIS 10.2. All the map and image data were projected into the UTM coordinate system Zone 45 using the WGS-84 geodetic datum. The accuracy of the manual digitization was controlled within one pixel.

### 2.2.2. Permafrost Data

The spatial distribution of permafrost type, the mean annual ground temperature (MAGT) of the permafrost and ground ice content, the soil temperature for the active layer and the ground temperature of the permafrost for boreholes at the observation sites along the Qinghai–Tibet Highway were provided by the Cryosphere Research Station on the Qinghai–Tibet Plateau, Chinese Academy of Sciences. Soil temperatures were monitored using 105 T thermistor sensors, with an accuracy of 0.1 °C. Instruments were attached to a CR1000 data logger (Campbell Scientific). Soil temperatures for the active layer were collected once every 30 min or 2 h (once every 30 min for QT observation fields; once every 2 h for CN observation fields) and permafrost temperatures for boreholes were automatically recorded 12 times per day at 2 h intervals. The ALT was determined by measuring the maximum depth of the 0 °C isotherm, observed from the soil temperature profile. The ALT at the four active layer observation sites (Table 1), the long-term averaged ALT during 1981–2017 calculated using the active-layer monitoring data recorded at more than ten observation sites and a long sequence of meteorological variables using a multiple regression method, as well as the ground temperatures at 15 m depth at the four borehole observation sites (Table 2) in this paper, were used to analyze the change in permafrost characteristics.

**Table 1.** The geographic information of the active layer observation sites.

| Location | Station Number | Longitude (°E) | Latitude (°N) | Altitude (m) | Underlying Surface Type |
|---|---|---|---|---|---|
| Kunlun pass | CN06 | 94.07 | 35.62 | 4746 | Alpine frost sparse steppe |
| Fenghuoshan | CN01 | 92.90 | 34.73 | 4896 | Alpine frost meadow |
| Kaixinling | QT05 | 92.40 | 33.95 | 4652 | Alpine frost desert steppe |
| Liangdaohe | CN04 | 91.73 | 31.82 | 4808 | Alpine frost marsh meadow |

Note: The second column is the station number, which comes from the Cryosphere Research Station on the Qinghai–Tibet Plateau, Chinese Academy of Sciences.

**Table 2.** The geographic information of the borehole observation sites.

| Location | Station Number | Longitude (°E) | Latitude (°N) | Altitude (m) | Underlying Surface Type |
|---|---|---|---|---|---|
| Xidatan | QTB01 | 94.08 | 35.71 | 4530 | Alpine frost sparse steppe |
| Hoh Xil Bridge | QTB06 | 93.26 | 35.29 | 4563 | Alpine frost sparse steppe |
| Wenquan | QTB15 | 91.89 | 33.09 | 4960 | Alpine frost sparse steppe |
| Liangdaohe | QTB18 | 91.73 | 31.81 | 4808 | Alpine frost meadow |

Note: The second column is the station number, which comes from the Cryosphere Research Station on the Qinghai–Tibet Plateau, Chinese Academy of Sciences.

### 2.2.3. Calculation of Ground Ice Content and Its Change

The ground ice in the permafrost was mainly distributed at a 1~10 m depth below the permafrost table, in particular, it was enriched at the 0~1 m depth below the permafrost table. The ground ice content was calculated by the formula: $Gi = \int \rho_d(Z)\theta(Z)dzDs$, where $Gi$ was ground ice content (kg), $\rho_d(Z)$ was bulk density of the soil (kg·m$^{-3}$), $\theta(Z)$ was gravimetric water content of the soil (%), $Z$ was permafrost thickness (m) and S was permafrost area (m$^2$). Based on the parameters from the borehole drilling and permafrost area of $1.06 \times 10^6$ km$^2$, the calculated ground ice content in permafrost on the QTP was 12,700 km$^3$.

Actually, the ground ice content decreased with permafrost degradation, but it was hard to evaluate the spatial change in ground ice content over the endorheic basin since it was difficult to obtain the water content change in the deep permafrost. Under the consequence of permafrost thawing, the permafrost table migrated downward and the ground ice has melted; thus, the increase in ALT

can be used to assess the change in thickness of ground ice. The X-G algorithm proposed by Xie and William [42] was used to accurately simulate ALT. This algorithm provided a new, simple algorithm that applied the Stefan equation to calculate the freezing (or thawing) depth in a multi-layered soil. It avoided the pitfalls of averaging the soil parameters in a multi-layered soil and resulted in a small systematic error between the observations and simulations [42,43]. The input thermal parameters included thermal conductivity, bulk density, gravimetric water content, N-factor and degree-days of thawing. Degree-days of thawing were estimated from the daily air temperature recorded at 87 weather stations of the National Meteorological Information Center (NMIC), China Meteorological Administration (CMA) (http://cdc.cma.gov.cn), and its spatial distribution was obtained by linear interpolation. Vegetation distribution of the permafrost zone on the QTP was produced by Wang et al. [44], and the N-factor was determined by the measured value for different vegetation types along the Qinghai–Tibet Highway. Soil data was from the 1:1,000,000 digital Second National Soil Survey of China from the Harmonized World Soil Database version 1.1 completed by the Institute of Soil Science, Chinese Academy of Sciences, Nanjing (http://westdc.westgis.ac.cn), which included the content of gravel, sand, silt and clay within topsoil (0~30 cm) and subsoil (30~100 cm). Soil density, water content and thermal conductivity were derived from the borehole data collected during field investigation (Table 3). In particular, the averaged unfrozen water content was 5% based on the experimental result of 4%~5% for sandy soil at different temperatures [45].

**Table 3.** The thermal parameters for the different soil types.

| Soil Type | Thermal Conductivity (W·m$^{-1}$·°C$^{-1}$) | Bulk Density (kg·m$^{-3}$) | Gravimetric Water Content (%) |
|---|---|---|---|
| Sand | 1.42 | 1800 | 6 |
| Silt | 1.575 | 1400 | 16 |

## 3. Results

### 3.1. Lake Expansion Pattern

The statistical results for the lake area and number in 2017 showed that there was a total of 434 lakes (>5 km$^2$) with a total area of 33,974.85 km$^2$ in the endorheic basin. The total lake area showed an overall increasing trend during 2000–2017; the total lake area was 27,281.10 km$^2$ in 2000 and 33,974.85 km$^2$ in 2017 with a significant increase of 6693.74 km$^2$ at a rate of 348.45 km$^2$/yr. The rate of area increase before 2010 (402 km$^2$/yr) was much larger than that after 2010 (313 km$^2$/yr) (Figure 2). Most lakes (323 lakes) in the endorheic basin were expanding dramatically from 2000 to 2017, with a significant area increase of 0.19~465.32 km$^2$ at rates of 0.01~26.29 km$^2$/yr. Only a few lakes (36 lakes) were shrinking, with slight area decrease of 0.06~96.08 km$^2$ at rates of 0.01~7.80 km$^2$/yr (Figure 3). Four lakes with an area increase greater than 200 km$^2$, Seling Co, Ayakekumu Lake, Duogecorenqiang Co and Aqikekule Lake were observed, with an increase of 465.32 km$^2$, 366.33 km$^2$, 215.70 km$^2$ and 208.36 km$^2$ at rates of 26.29 km$^2$/yr, 22.18 km$^2$/yr, 10.64 km$^2$/yr and 11.78 km$^2$/yr, respectively. The spatial pattern of lake expansion presented a southwest–northeast transition. Dramatic lake expansion has mainly occurred in six centralized subregions in the northern and eastern endorheic basin: the West Kunlun Mountains and Karakoram Mountains (I), the Altun Mountains (II), the western Hoh Xil region (III), the central eastern Hoh Xil region (IV), the northern slope of Tanggula Mountains (V) and Seling Co and its surrounding large lakes (VI). The most significant lake expansion has occurred in the Hoh Xil region (III and IV) (Figure 3a).

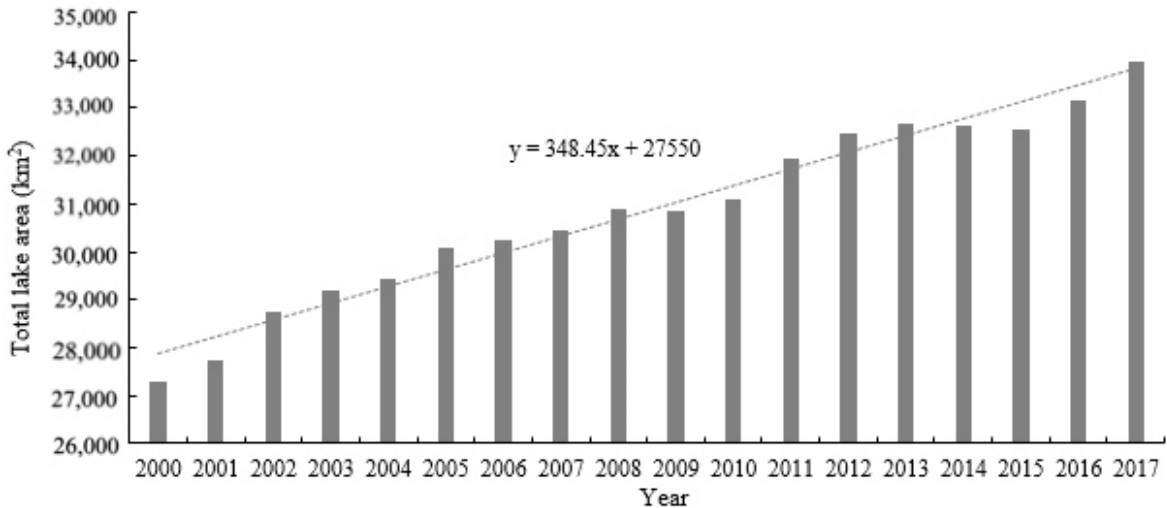

**Figure 2.** Annual change in total lake area in the endorheic basin from 2000 to 2017.

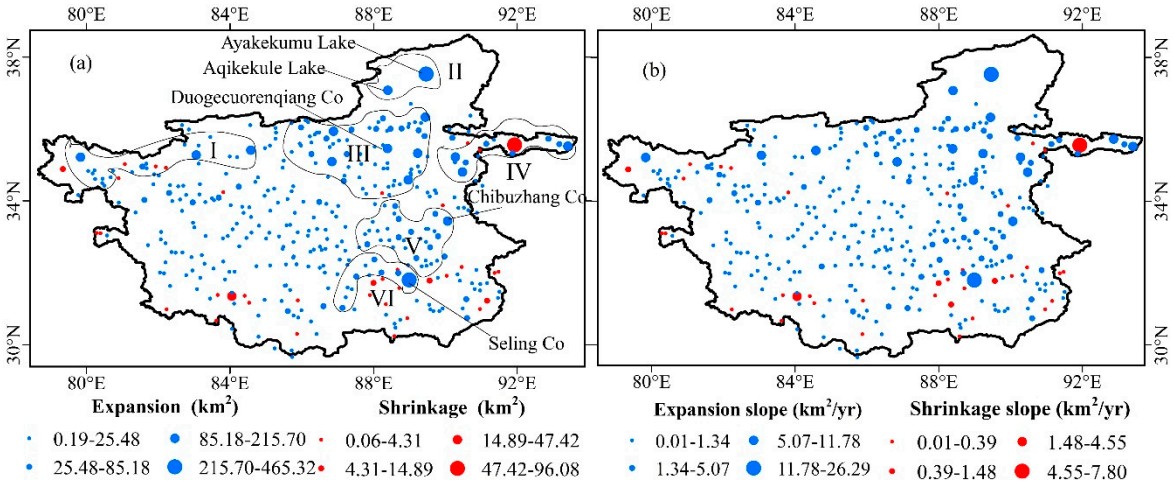

**Figure 3.** Spatial patterns of trends in lake-area change (**a**) and change rate (**b**) from 2000 to 2017 in the endorheic basin. The black line contour is the Qinghai–Tibet Plateau (QTP) endorheic basin; the blue dots mean lake expansion and red dots mean lake shrinkage.

### 3.2. Lake Expansion and Permafrost Characteristics

### 3.2.1. Permafrost Types

The permafrost area over the endorheic basin was $0.42 \times 10^6$ km$^2$, accounting for 60% of the total area over the endorheic basin and 40% of the total permafrost area on the QTP. The continuous permafrost distributed in the northern and eastern endorheic basin and the island-discontinuous permafrost distributed in the western–southern endorheic basin with small area. The spatial pattern of lake expansion in the endorheic basin from 2000 to 2017 showed a southwest–northeast transition from shrinking to stable to rapidly expanding, which corresponded well with the permafrost distribution from island-discontinuous to seasonally frozen ground to continuous permafrost (Figure 4a). A dramatic lake expansion was observed in the continuous permafrost; 241 lakes (74.6%) of the total 323 expanding lakes were distributed in the continuous permafrost, and the area increased by 5273.85 km$^2$, accounting for 77% of the total area increase (6842.74 km$^2$). In addition, 42 of 48 typically expanding lakes with an area increase greater than 30 km$^2$ were distributed in continuous permafrost. In addition, lake expansion in the continuous permafrost showed a large spatial difference (Figure 4b). The lakes expanded slightly in the southern continuous permafrost (the southern boundary of continuous permafrost on the QTP), while lakes expanded significantly in the northern and eastern continuous

permafrost (the hinterland of continuous permafrost on the QTP); 33 of the 42 typically expanding lakes, 15 expanding lakes with area increase greater than 100 km$^2$ and 3 expanding lakes with area increase more than 200 km$^2$ in continuous permafrost were located in the northern and eastern continuous permafrost. A significant lake expansion in the continuous permafrost stood in sharp contrast with the lake shrinkage in the southern zones of the island-discontinuous permafrost; 23 (63.8%) of the 36 shrinking lakes were distributed in island-discontinuous permafrost in the southwestern endorheic basin (Figure 4a).

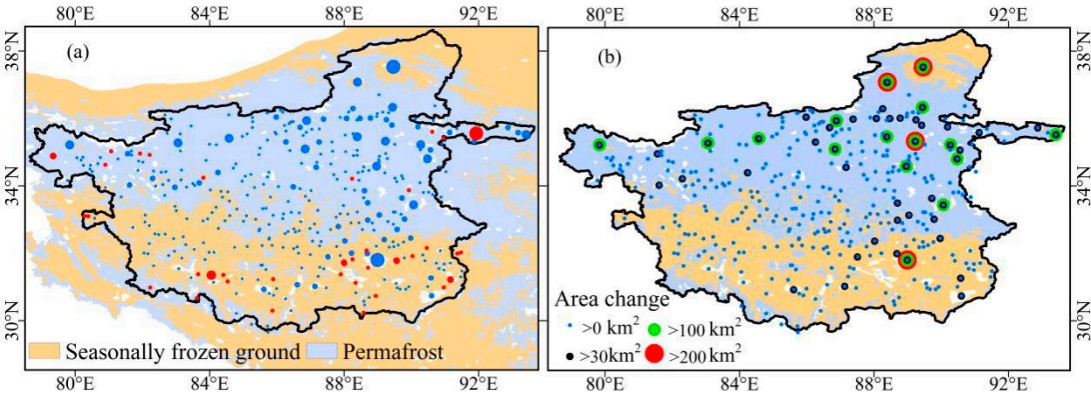

**Figure 4.** (**a**) The spatial patterns of lake change and permafrost types over the endorheic basin (the blue dots mean lake expansion and red dots mean lake shrinkage); and (**b**) the spatial difference of lake expansion in continuous permafrost over the endorheic basin. The permafrost distribution was provided by Zou et al. [1]. The black line contour is the QTP endorheic basin.

### 3.2.2. Ground Ice Content

According to the MAGT of permafrost, permafrost on the QTP was classified into extremely stable permafrost (MAGT < −5.0 °C), stable permafrost (−5.0 °C < MAGT < −3.0 °C), metastable permafrost (−3.0 °C < MAGT < −1.5 °C), transitional permafrost (−1.5 °C < MAGT < −0.5 °C) and unstable permafrost (−0.5 °C < MAGT< 0.5 °C) [46]. Permafrost was also classified into warm permafrost (MAGT > −1.5 °C) and cold permafrost (MAGT < −1.5 °C) [47]. The spatial characteristics of the MAGT in the endorheic basin showed that the stable and sub-stable cold permafrost were distributed in the northern and eastern continuous permafrost and the transitional and unstable warm permafrost distributed in the southern continuous permafrost (Figure 5). The MAGT was taken as a significant factor that influenced ground ice content in permafrost. The spatial distribution of the ground ice content in the endorheic basin was shown in Figure 6. The total ground ice content over the endorheic basin was 6157 km$^3$ with a range of 0~222 × 10$^6$ m$^3$, accounting for 50% of the total amount of ground ice on the QTP. The spatial pattern of the lake expansion in the continuous permafrost agreed well with the spatial distribution of ground ice content that mainly depended on the MAGT and permafrost thickness; the lake expanded significantly in the northern and eastern continuous cold permafrost with a large amount of ground ice content, whereas the lake expanded slightly in the southern continuous warm permafrost with a low ground ice content (Figure 6).

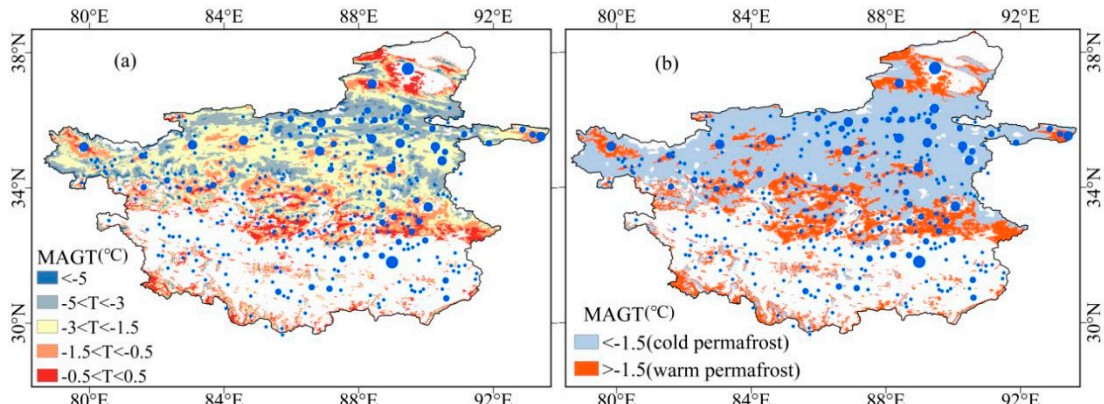

**Figure 5.** (**a**) The spatial patterns of lake expansion and the MAGT of the permafrost over the endorheic basin; and (**b**) the spatial difference of lake expansion in warm and cold permafrost over the endorheic basin. The black line contour is the QTP endorheic basin.

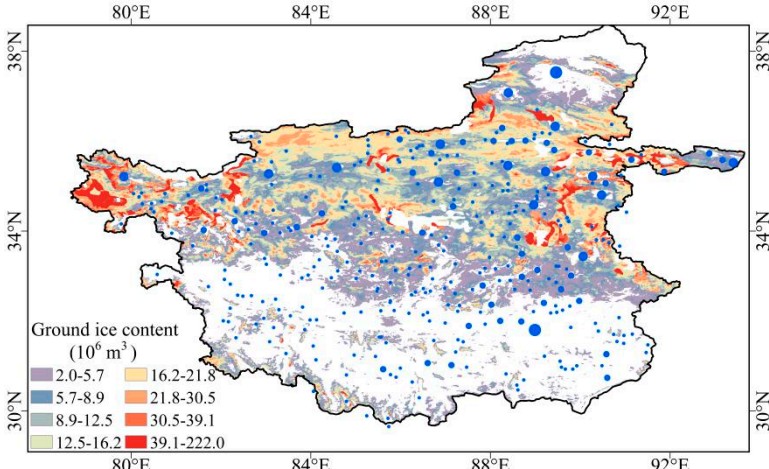

**Figure 6.** The spatial patterns of lake expansion and ground ice content over the endorheic basin. The black line contour is the QTP endorheic basin.

*3.3. Permafrost Degradation Evidence*

The long-term averaged ALT along the Qinghai–Tibet Highway experienced an overall increasing trend, varying from 181 cm in 1981 to 238 cm in 2017, increasing by 57 cm at rate of 1.96 cm/yr (Figure 7a), which was larger than the increase rate of ALT (1.29 cm/yr) on the whole QTP produced from Xu et al. [24] and that (1.33 cm/yr) along the Qinghai–Tibet Highway during 1981–2010 from Li et al. [25]. The ALT since 2000 showed a rapid increase rate of 2.60 cm/yr with an averaged value of 218 cm, which was approximately double the increase rate of 1.44 cm/yr before 2000 with a mean value of 184 cm. The ALTs at observation sites along the Qinghai–Tibet Highway also showed rapid increasing tendencies; the ALTs increased by 30 cm at a rate of 2.42 cm/yr during 2004–2017 at the Kunlun pass site (CN06), 64 cm at a rate of 4.26 cm/yr during 2000–2017 at the Fenghuoshan site (CN01), 31 cm at a rate of 3.17 cm/yr during 2004–2017 at the Kaixinling site (QT05) and 32 cm at rate of 1.59 cm/yr during 2000–2017 at the Liangdaohe site (CN04), respectively (Figure 7b). The mean ALT varied from 112 cm at the Liangdaohe site to 304 cm at the Kaixinling site, with an average of 185 cm from four sites over the period of their records. Our results suggested that the permafrost degradation has occurred in the endorheic basin on the QTP in recent years, especially the accelerated permafrost degradation since 2000.

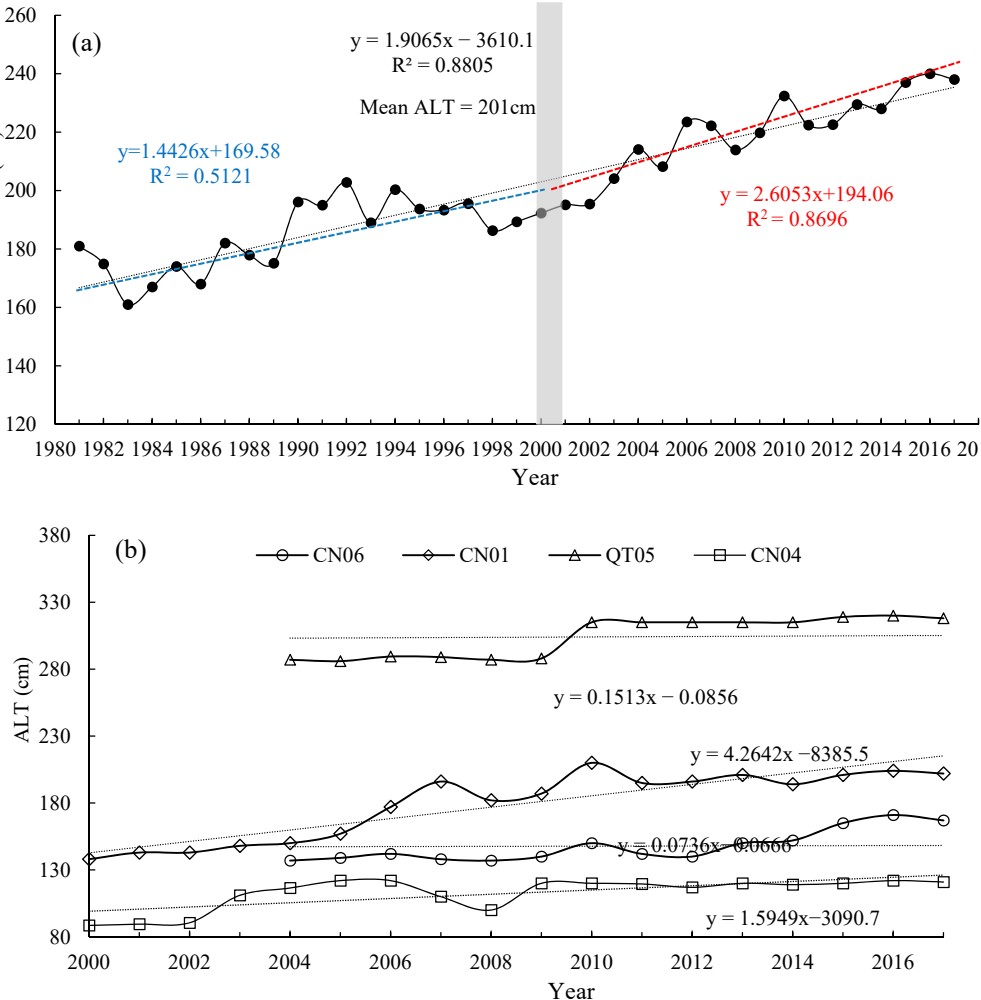

**Figure 7.** The annual change in averaged active layer thickness (ALT) during 1981–2017 (**a**) and the ALTs at four observation sites (**b**) along the Qinghai–Tibet Highway.

In addition, soil temperatures at 15 m depth from the borehole sites were used for describing permafrost temperatures in this study, because the depth of the zero amplitude ranged from 10 to 15 m depth on the QTP and the permafrost temperatures at 15 m depth on the QTP were generally within −2°C from the freezing point, except for a few mountain areas, thus avoiding any seasonal temperature variation effect [48]. The soil temperatures at 15 m depth at the observation sites along the Qinghai–Tibet Highway were also rising slightly, with rates of 0.0054 °C/yr during 2004–2017 at Xidatan site (QTB01), 0.0028 °C/yr during 2005–2017 at Hoh Xil Bridge site (QTB06), 0.0346 °C/yr during 2005–2017 at Wenquan site (QTB15) and 0.0252 °C/yr during 2004–2017 at Liangdaohe site (QTB18), with an average of 0.017 °C /yr from the four sites (Figure 8). The mean soil temperatures at 15 m depth at four sites were −0.25 °C, −0.52 °C, −0.88 °C and −0.66 °C, respectively, with the highest value at the Xidatan site (QTB01) and lowest value at the Wenquan site (QTB15), which was consistent with the trend of MAGT of −0.45 °C, −0.47 °C, −1.03 °C and −0.73 °C at the four sites, respectively [28]. Permafrost at the four sites were warm permafrost, and the magnitude of the ALT increase of the warm permafrost was greater than that of the cold permafrost, but the magnitude of the permafrost temperature rise was less than that of the cold permafrost [48]. As described above, most of the regions in the endorheic basin were covered by cold permafrost (Figure 5b); its ground temperatures were supposed to increase much more than the warm permafrost along the Qinghai–Tibet Highway. The above results further indicated that permafrost in the endorheic basin on the QTP was warming, thawing and degrading inevitably.

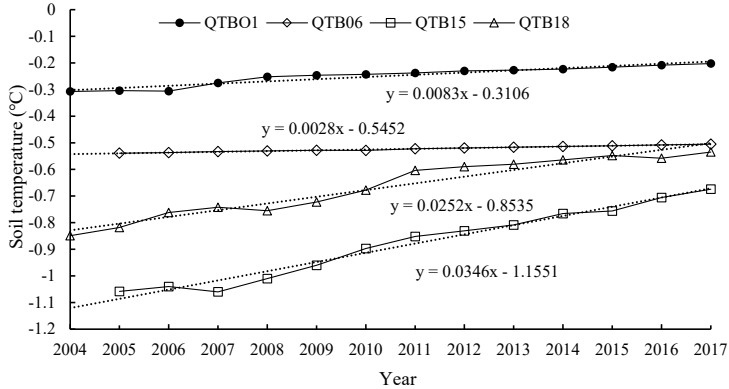

**Figure 8.** The change in soil temperature at 15 m depth of permafrost at four observation sites along the Qinghai–Tibet Highway.

### 3.4. Contribution of Permafrost Degradation to Lake Expansion

The simulated ALT was verified by measured data (Figure 9). The measured ALTs at the Zonag Lake, Yanhu Lake and Longmu Co sites were 2.0 m, 2.2 m and 2.5 m, respectively, which were close to the simulated values of 1.78 m, 2.03 m and 2.69 m, with relative errors of 11%, 7% and 7%, respectively, which indicated that the simulated ALT can reveal the change trend in ALT over the endorheic basin. The simulated averaged increase in ALT in the endorheic basin from 2000 to 2016 was 0.255 m with a range of −0.90~0.73 m (Figure 10). The averaged soil water content at the permafrost table from many field sites along the Qinghai–Tibet Highway was 27%. The estimated ground ice content below the permafrost table over the endorheic basin decreased by 26.02 km$^3$ from 2000 to 2016. Yang et al. [12] analyzed the variation in volume of closed lakes larger than 50 km$^2$ on the QTP; the total lake volume exhibited a net increase of 99.64 km$^3$ at rate of 7.67 km$^3$/yr during 2000–2013, which was consistent with the significant increase rate of lake volume (7.72 ± 0.63 km$^3$/yr) in the endorheic basin during 2003–2009 [13]. Thus, the lake volume increased by 122.65 km$^3$ from 2000–2016 in the endorheic basin, and the contribution of ground ice melting to the total increase in lake volume was 21.2%. Meanwhile, according to the rate of ground ice meltwater in the Dongkemadi River Basin, Tanggula Pass, on the QTP $(0.0396 \times 10^5$ m$^3$/km$^2)$ [20], the annual meltwater from ground ice in the endorheic basin within a permafrost area of $0.42 \times 10^6$ km$^2$ was 1.6632 km$^3$, which accounted for 21.6% of the annual increase in lake volume.

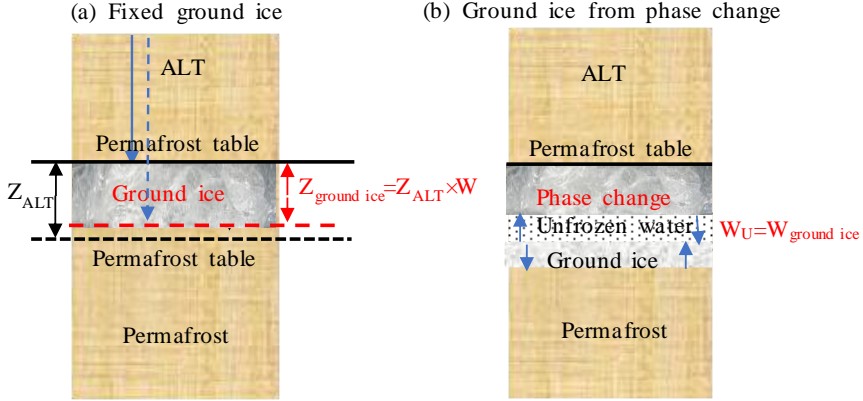

**Figure 9.** The influential mechanism of permafrost thawing on lake change. $Z_{ALT}$ is the increased thickness of the ALT; W is the soil water content at the permafrost table; $Z_{ground\ ice}$ is the melt thickness of the ground ice; $W_u$ is the increase in the unfrozen water content; $W_{ground\ ice}$ is the decrease in water content due to the phase change during the freezing–thawing process of active layer.

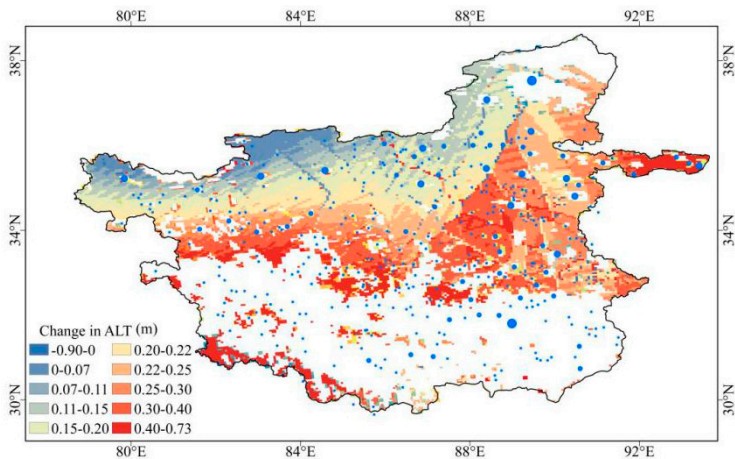

**Figure 10.** The spatial patterns of lake expansion and distribution of change in ALT during 2000–2016 over the endorheic basin. The positive numbers show an increase in the ALT and the negative numbers a decrease in the ALT; the black line contour is the QTP endorheic basin.

In addition, a certain amount of unfrozen water was widely distributed in the permafrost, and the increase in unfrozen water content associated with the rising ground temperature of the permafrost will provide free water during the thawing–freezing process (Figure 9a). A simple power law was usually used to relate ground temperature T (°C) and gravimetric unfrozen water content Wu (Wu = a$|$T$|^{-b}$, where a and b were empirical constants, which was soil parameters for different soil types) [45]. The unfrozen water content increased exponentially with permafrost ground temperature, and it reached the maximum when the ground temperature approached 0 °C (freezing point). During 2004–2017, the average annual ground temperature at 15 m depth from the QTB01, QTB06, QTB15 and QTB18 observation sites showed a rapid rising trend, varying from −0.87 °C in 2004 to −0.47 °C in 2017 at rate of 0.025 °C/yr. Accordingly, the unfrozen water content increased with the rising permafrost ground temperature for the same period, ranging from 0.78% to 1.13% at a rate of 0.0216%/yr (Figure 11). The increase in unfrozen water content implied that the decrease in ground ice content, which participated in phase change in permafrost during the freezing–thawing process of the active layer. The estimated annual ground ice content decreased by 1.32 km$^3$, which contributed 17.2% for the annual increase in lake volume. Overall, the ground ice melting due to permafrost degradation has a total contribution of 38.4% for the lake volume increase. Compared with the result proposed by Zhang et al. [13], ground ice melting due to permafrost degradation indicated a water release of 0.92 ± 0.46 km$^3$/yr, which contributed 12% for the lake volume increase in the QTP's endorheic basin; this result was far less than our result because the released water was estimated using the active layer depth model from Xiang et al. [49], with a range of −2.4~0.3 mm/yr on the QTP, which was less than previous results of −0.059~0.1013 m/yr (averaged value of 0.0129 m/yr during 1981–2010) from Xu et al. [24] and the averaged ALT increase rate of 0.31 m/yr during 1981–2013 conducted by Qin et al. [23].

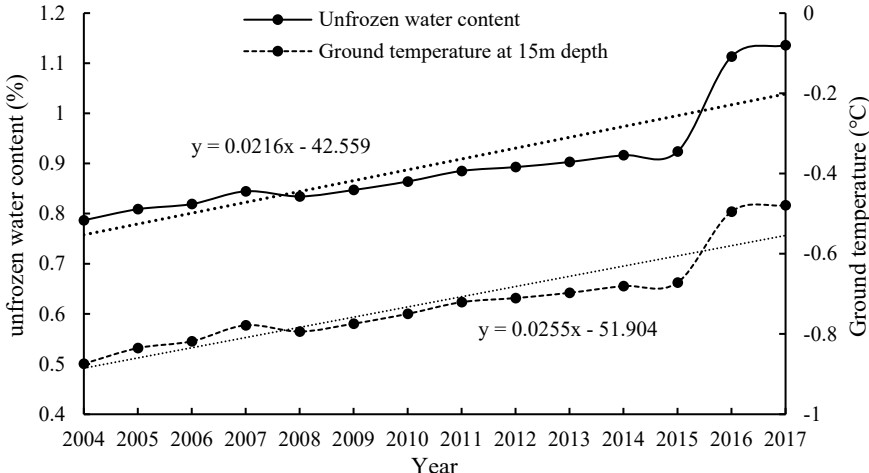

**Figure 11.** The change in unfrozen water content with the ground temperature of the permafrost.

## 4. Discussion

### 4.1. Cause of Lake Expansion and Shrinkage in Permafrost

The phenomenon of lake expansion in continuous permafrost but lake shrinkage in island-discontinuous permafrost was also observed in the River Source Region [50] and on the Mongolian Plateau [51,52]. The lake expansion in continuous permafrost was mainly because of the presence of ice-rich permafrost with high permafrost coverage serving as a barrier layer due to low hydraulic conductivity and permeability, which has impeded the liquid water infiltration and the interaction between the surface water and groundwater, and finally resulted in a larger direct surface runoff (runoff ratio) in both rain and snow-melt and a lack of a water storage buffer effect than those in non-permafrost regions. In general, catchments covered by a higher permafrost extent could have lower groundwater storage capacity and thus a higher summer peak flow and a lower winter base flow [53–55]. Plus, the fact that the potential evaporation in continuous permafrost was smaller due to the cooler climates than that in the island-discontinuous permafrost [34], suggested there is a lower MAGT of continuous permafrost than that of island-discontinuous permafrost (Figure 5). This phenomenon was also due to more meltwater from ground ice associated with permafrost degradation. As the ground ice began to melt, the permeability of the soil increased and the hydraulic exchanges between the surface water and suprapermafrost groundwater was enhanced. Thus, aquifer activation arising from permafrost degradation may increase the recharge and groundwater discharge to supply rivers and lakes.

Whereas island-discontinuous permafrost was unable to form stable aquicludes because of a small area and thin thickness, the response to permafrost degradation was presented as taliks expansion, which allowed subsurface drainage. Maybe, as mentioned by Smith et al. [56] and Riordan et al. [57], initial permafrost thawing led to lake expansion and was followed by a declining water surface as drainage due to further permafrost degradation. Shrinking lakes may become a common feature in island-discontinuous permafrost; permafrost degradation was also linked to the rapid shrinking thermokarst ponds in the Arctic region [56–58], and a dropping groundwater table and lowering lake water levels at the source areas of the Yangtze River and Yellow River were closely related to accelerated permafrost degradation [59]. Permafrost degradation through taliks provides a major pathway to connect surface water and groundwater systems; the talik or talik channel has an enlarged or penetrating talik (open taliks), and has been formed from accelerating permafrost degradation, so that a hydraulic connection was established between the surface water and groundwater. This has facilitated a large amount of surface water and suprapermafrost groundwater discharge to subpermafrost groundwater or subpermafrost aquifers via taliks, and correspondingly induced declining groundwater levels and thereby shrinking lakes. Meanwhile, the groundwater in island-discontinuous permafrost and

its alongside neighbors was connected and transformed mutually, and the increased recharge and discharge supplied groundwater in the adjacent neighbors at low heads with low elevation through taliks (activated aquifers), which has caused lake expansion instead. This was manifested by the fact that lake shrinking has occurred in the southern Seling Co in island-discontinuous permafrost but lake expansion in the northern Seling Co in the continuous permafrost boundary.

### 4.2. Impact of Thawing–Freezing Process on Water Contribution

In theory, the meltwater from ground ice was a major water supply of permafrost degradation to lake water; the melting of ground ice provide more meltwater and increased soil permeability, which has enhanced the exchange between surface water and suprapermafrost groundwater, increased groundwater discharge to rivers and lakes and further contributed to lake expansion. In fact, the meltwater from permafrost degradation did not replenish lakes completely. Permafrost degradation in permafrost basins has been shown to smooth out the seasonal distribution of discharge at the catchment scale in a number of ways. The decrease in ground ice content in this study was estimated in later summer when the active layer reached its maximum and ground ice melt completely during the freezing–thawing process. Although the meltwater from ground ice released soil pore space to facilitate infiltration of surface water, and in some cases also elimination of soil and groundwater, this meltwater participated in the following autumn freezing process and some parts of them changed into ice, and the further freezing of infiltrated snow meltwater will form another layer of impervious ground ice. This state not only decreased the amount of meltwater that converted into groundwater effectively but also retarded the soil water and meltwater infiltrating and further increasing groundwater.

### 4.3. Impact of Permafrost Extent and Thermal Condition on Contribution

In addition, some of the meltwater that finally reached the groundwater directly influenced the groundwater recharge and lake water levels, or eventually drained off as spring water [60,61], or increased the groundwater discharge to surface drainage, as well as some of the meltwater that directly drained to become surface runoff, especially at a steep slope. However, whether this part of the meltwater can supply the lake completely was uncertain, because the effects of permafrost degradation to groundwater discharge and even surface runoff were highly depended on permafrost extent and the thermal characteristics of the watershed. Permafrost extent in one basin controlled the water distribution of surface–subsurface interactions; thus, basins with different permafrost extents have contrasting streamflow regime characteristics. The changes in annual discharge, the $Q_{max}/Q_{mi}$ and the baseflow were positively correlated with permafrost coverage. Song et al. [34] proposed that the more significant effects of permafrost coverage to the annual ratio of $Q_{max}/Q_{min}$ in high permafrost cover in QTP subbasins; $Q_{max}/Q_{min}$ ratios was low in the low to moderate (<50%) permafrost coverage basins, while moderate to high (50~100%) in the permafrost coverage basins. Ye et al. [38] described a relationship between the monthly ratio of $Q_{max}/Q_{min}$ and permafrost coverage in the Lena basin and suggested that permafrost condition does not significantly affect streamflow regime over the low permafrost (<40%) regions, and it strongly affects discharge regime for regions with high permafrost (>60%). These indicated the general applicability of such relationships, which existed in a larger spatial scale in crossregion permafrost basins with heterogeneity.

The endorheic basin on the QTP was divided into 281 subbasins, and most of them were covered by permafrost to a different extent. The permafrost coverage in 48 typically expanding lake subbasins were analyzed, and among 42 of the 48 typically expanding lakes in the continuous permafrost, three lakes that were located in the southern boundary of the continuous permafrost had a large increase in ALT but a low permafrost coverage less than 50% (27%, 39%, 49%), as well as low ground ice content, giving rise to less hydrological effects regarding groundwater and surface runoff, which was also another factor that overestimated the contribution of permafrost degradation to lake expansion. The remaining 39 lakes have high permafrost coverage in their basins with a range of 65%~99%, including 32 lakes with permafrost coverage more than 80% and 26 lakes with permafrost coverage

more than 80%. However, a weak positive correlation between lake area increase and permafrost coverage in their basins was observed; maybe this could be attributed to the permafrost thermal characteristics, since the rapid lake expansion was significantly due to precipitation increase.

Except for the deepening of the permafrost table and degradation of the ice-wedge systems, the thawing status of the sublacustrine permafrost was another potential permafrost-related driver of lake change, which also facilitates exchanges between the lake water and groundwater. In the cold regions, the hydrological regime was closely related with permafrost thermal conditions; the average status of the vadose zone in the low permafrost coverage basins was relatively steady under a warming climate, which has minor effects on discharge change with low permafrost coverage, while in the basins with high permafrost coverage, permafrost change occurs at a large scale under climate warming, which can lead to increased hydrological connectivity, increased infiltration and then change flow regime. There are substantial impacts of shallow permafrost thaw on lake water–groundwater exchange and hence lake water budgets. Jepsen et al. [62] suggested that the shallow (few tens of meters) thaw state of permafrost has more influence than deeper permafrost conditions on the evolving water budgets of lakes on a multidecadal time scale in interior Alaska, because shallow aquifers have higher hydraulic conductivity and greater spatial variability in the thaw state, which makes groundwater flow and the associated lake level evolution particularly more sensitive to climate change owing to the close proximity of these aquifers to the atmosphere. Wang et al. [53] also suggested that the active soil thawing in the upper layer of depth of 60 cm had contributed to an increase in discharge, but the increase in thawing depth deeper than 60 cm led to a decrease in surface runoff and slowness in the recession process.

Due to the different scales of permafrost coverage and thermal conditions in the different hydrological sections of a watershed over the endorheic basin on the QTP, the discussion on permafrost extent and changes in permafrost conditions is available, but it is difficult to determine the specific contribution for each lake basin, because the streamflow characteristics varied significantly in a basin. The effect of permafrost degradation on hydrological regime is complex, involving the potential interactions between climate change, permafrost degradation and groundwater flow. Thus, examining and comparing the hydrological regimes and various permafrost conditions and accurately quantifying the contribution of permafrost degradation to lake expansion in each lake basin is crucial and will be our next work.

## 5. Conclusions

The spatial-temporal change in lake area during 2000–2017 was examined and the influence of permafrost degradation on lake expansion in the endorheic basin on the QTP was discussed based on Landsat images and permafrost field data in this study. Permafrost characteristics and its degradation trend have close relationships wsith lake changes. Lake expansion in the endorheic basin showed a southwest–northeast transition from shrinking to stable to rapidly expanding, which corresponded well with the permafrost distribution from island-discontinuous to seasonally frozen ground to continuous permafrost. A dramatic lake expansion in the continuous permafrost showed significant spatial differences; lakes expanded significantly in the northern and eastern continuous permafrost with a higher ground ice content but slightly in the southern continuous permafrost with lower ground ice content. This spatial pattern was mainly attributed to the melting of ground ice in shallow permafrost associated with accelerating permafrost degradation. Whereas, some lakes in the southern zones of the discontinuous-isolated permafrost were shrinking, which was mainly because the extended taliks arising from intensified permafrost degradation has facilitated surface water and suprapermafrost groundwater discharge to subpermafrost groundwater, thereby draining the lakes. Based on observation and simulated data, the melting of ground ice at shallow depths below the permafrost table accounted for 21.2% of the increase in lake volume from 2000 to 2016.

**Author Contributions:** Conceptualization, W.L.; software, H.L.; investigation, H.L.; data curation, W.W., G.Y., Y.Z., G.L., Q.P. and D.Z.; writing—original draft preparation, W.L.; supervision, C.X. and T.W.; project administration, X.X. All authors have read and agreed to the published version of the manuscript

**Funding:** This study was supported by the Strategic Priority Research Program of Chinese Academy of Sciences (Nos. XDA23060703), the National Natural Science Foundation of China (No. 41671068), the State Key Laboratory of Cryosphere Sciences (No. SKLCS-ZZ-2019) and The Second Tibetan Plateau Scientific Expedition and Research program (No. 2019QZKK0905).

**Conflicts of Interest:** The authors declare no conflict of interest.

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
