# Peer review of "The Impact of Permafrost Degradation on Lake Changes in the Endorheic Basin on the Qinghai–Tibet Plateau"

_water, doi:10.3390/w12051287_

Round 1

Reviewer 1 Report

General comments "The impact of permafrost degradation on lake changes in the endorheic basin on the Qinghai-Tibet Plateau".

The research describes the impact of permafrost degradation on sizes of lakes in the endorheic basin on the Qinghai-Tibet plateau over the period of 2000-2017. The research explores what is driving the differences in sizes of lakes and uses GIS data to asses the sizes and models to assess the permafrost degradation and ground ice. This research is of high importance, since it combines different sources of data and integrates this in a tool that describes the dynamics of permafrost and size of lakes. I was surprised to read that many lakes have a very rapid increase in size, up to 26 km2/yr.

However I feel that the manuscript needs significant restructuring and rewriting. Many sentences are way to long and are therefore very complex, also the level of English is mediocre. I suggest to send it to a proofreader after making the following corrections:

1) the term island-discontinuous and discontinuous isolated are intermingled.

2) Please present a map where all field stations are located, based on your study area, and also indicate elevation on this map. In this way the reader get a better feeling with the research area (probably this is figure 1, but figure 1 is missing). I can therefore not review if the study is scientifically sound, since I have no idea where the field stations from which the data is coming from are located. 

3) Use the correct name of companies, for example Campbell scientific

4) Describe the resolution of LandSat.

5) in section 3.2.2. methods are introduced. This should not be in the result section.

6) line 11 on page 10: Here it is written that the model was calibrated, however more information is needed how this was done and on what data.  In a scientific manuscript the reproducibility is of high importance.

The manuscript has different fonts and font sizes, giving the idea that this manuscript is was a hasty job and not taken seriously.

My conclusion about this manuscript is that it needs significant restructuring and grammar corrections. When this and the points above have been addressed, the authors should resubmit the manuscript and go through another review process.

Reviewer 2 Report

Major

This study shows the increase in the lake area in the continuous permafrost zone in QTP. The authors hypothesize that melted ground ice feeds the lakes contributing to the lake expansion. The distribution of the ground is the most critical factor that would define the lake expansion. The algorithm of the ground ice distribution needs to be better explained. The associated with ground ice uncertainties need to be well presented. The volume of the ground ice should be related to the volume of water in the expanded lakes.  How could authors be so sure that it is not groundwater pathways that are not related to the ground ice?

I am not convinced that the amount of active layer ground ice is enough to contribute to lake expansion? It is possible that the shrinking of the nearby lake could contribute to the other lake expansion.

The increase of the active layer is not enough to account for the melted ground ice. For example, Lui et al. (2014) and Schaefer et al. (2015) used InSAR data to calculate the subsidence in Alaska. Note, that in the Schaefer's study area, the active layer did not increase; instead, it was balanced by the ground subsidence. If authors provide subsidence maps for some of the areas within the study area, then it will be more convincing to relate subsidence to the melted ground ice. Once the authors add the subsidence map to the study, I will be more willing to recommend this paper for the publication.

Minor

L8. statistics

No figure 1

Section 2.2.2. … sue data?

Why was the Stefan model used? How this approach compares with the other empirical, analytical methods. 

Figure 10. Legend in chinses

The captions for figures need to be expanded. So that they can independently explain the Figures. 

The English grammar needs to be check by the professional.

Reference

Schaefer, K.M., Liu, L., Parsekian A., Jafarov E.E., Chen, A., Zhang T., Gusmeroli A., Zebker,. H., Schaefer. T.: (2015), Remotely Sensed Active Layer Thickness (ReSALT) at Barrow, Alaska Using Interferometric Synthetic Aperture Radar. Remote Sensing 7, no. 4 (2015): 3735-3759.

Liu, L., Jafarov E.E., Schaefer K.M., Jones B. M., Zebker H.A., Williams C.A., Rogan J., Zhang, T.: (2014), InSAR detects increase in surface subsidence caused by an Arctic tundra fire. GRL. doi: 10.1002/2014GL060533

Reviewer 3 Report

Below you can find some of my suggestions by page and line, followed by some general ones, more related to format, typos, and language. As a not English native speaker, my language recommendations and edits should be considered accordingly.

  • [Page 2 Line 33] missing an r “on river runoff”
  • [Page 3] Figure 1 is missing.
  • [Page 3] Some images of the process of extracting the lake area could be interesting. I would like to see an example of how the lakes look on the two Landsat color compositions and how they were later delineated.
  • [Page 4 Line 4] “The accuracy of manual digitization was controlled within one pixel” I understand that this means that when delineating, lakes polygons were edited to the pixel scale. But it could be good to include the resolution of that pixel in order to know the exact accuracy and how it influences the final area
  • [Page 4 Line 5] “2.2.2. Permafrost in sue data” Not sure of the meaning of “sue data”, do you mean “in-situ”? Also, I am not sure I understand where the data came from. There are several observation sites (in 3.3. it is stated that there are “more than twenty observation sites” but here “more than ten”), but only 4 and 4 are listed in tables 1 and 2. Are they grouped around these sites? Are they more, but not listed? Why more than 20, and not an exact number? Why these observation sites are not explained in this section but used in the result section? It is a bit confusing for me and I believe it could be better explained.
  • [Page 4 Line 12] “collected once every 30min or 2h” Why these differences? Those different data collection ranges are for different observation sites? Or a site has different data collection times over the years? Please, explain this better. If it is the first case, Include the data frequency in Tables 1 and 2.
  • [Page 4 Table 1 and 2] I am having problems locating these sites owing to the lack of Figure 1, since I assume there will be indicated there. If that is the case, cite here the figure. If not, it is advisable to include these sites in the schematic map, for reference. Also, the second column is not a number but a reference/identification. If this nomenclature comes from the Cryosphere Research Station, it should be clarified. Are there more sites? Why these were the ones selected? And why some are active layer and other borehole observation sites? Is because the later are deeper (>15m depth)?
  • [Page 4 Lines 22 to 24] Rephrase. Too long and confusing sentence.
  • [Page 4 Line 29] Misspelled. Change “by linear interpolation.” to “by linear interpolation.”
  • [Page 5 Lines 17 to 19] This sentence is redundant with the upcoming text. It is advisable to rephrase or delete it.
  • [Page 5 Lines 20, 21 and 22; Page 7 Line 31] I understand that 0.19~465.32km2 is an interval. The use of “~” for intervals could be misleading, I advise the use of “-“.
  • [Page 6 Figure 3] Caption are self-explanatory, all need to include all the information to understand the figure by themselves. Include in all captions with maps the meaning of the black line contour (The QTP endorheic basin, I am guessing). Texts are not complete, and the lower half is cut. If you could only include degrees in the coordinates, it would look cleaner. Also, I see two expansion lakes not included in any area. Those areas are geographic, but they can be adjusted to include them? (between area I and II). Finally, “Expansion and Shrinkage slopes” are “rates” instead, right?
  • [Page 7 Figure 4] Caption needs to include the meaning of the black line contour but also of the dots. They are not the same for (a) and (b), you can call Figure 3. Also, an explanation is needed for what could be seen in parts (a) and (b).
  • [Page 8 Figure 5] Similar problems to Figure 4. Also (a) and (b) are somewhat redundant. Are not they the same data but with different reclassification? Yellow and blue in (a) is the same thing than blue in (b); and reds in (a) is red in (b). Not sure if these two parts are needed. Maybe you can indicate the cold and warm permafrost threshold in the legend, and use (b) for figure 6.
  • [Page 8 Line 12] Why do you separate the data before and after the year 2000? Why don’t you look to the same range of years as Li et al. to make a direct comparison with their results?
  • [Page 9 Figure 7] Axis's name in (b) is overlapping the numbers. Both axis labels could be sparser for clarity (each 50 cm, every 2 or 5 years). The year data is the average from January 1st to December 31st? I am assuming this but should be clarified both in the caption and the text (either here or in the 2.2.2 section).
  • [Page 9] I am yet confused by the source of the data from (a) and (b), but I find interesting that the ALT for 3 of the 4 sites used in (b) is way lower than the mean in (a) for the same period. Also, QT05 (Kaixinling, that also has a different notation than the other 3, for some not explained reason) is way higher. That makes me wonder if there is high variability in those “more than twenty sites” showed in (a) so maybe a range and/or standard variation for those ALT values could provide useful information.
  • [Page 10 Line 23] “7.72±0.63Gt/yr” Why use gigatons when km3 is the volume unit used in the text?
  • [Page 11 Figure 9] (a) and (b) text are duplicated on both sides. The dashed line around the two diagrams as well as the greenish squares might not need and could be deleted for simplification and clarity.
  • [Page 11 Figure 10] Legend is in Chinese. Same recommendation than for the other maps.
  • [Page12] I am not familiar with “discontinuous-island permafrost”. To me, just discontinuous permafrost is clear enough.
  • [Page 13] The use of “enhance” does not sounds right to me. Maybe increase recharge/discharge?
  • [Page 13 Line 34] “these meltwater” The water is uncountable, use “this meltwater”
  • [Page 14 Line 1] “4.3. Impact of permafrost extent and thermal condition on contribution” It sound incomplete. Maybe “on water contribution”?
  • [Page 15 Lines 4 and 9] I am not sure what “earth lake basin” and “earth subbasins” are. What do you mean with that? please clarify.
  • [Page 15 Line 9] “was crucial and was our next work.”, do you mean that “is crucial” and “will be our next work”?
  •  

Minor formatting comments:

  • Differences in font type and size during the text, misplaced italics (as in “the” line 42). It is not an actual problem for publication, this will be easily fixed, but it difficult the review.
  • Double spaces in [Page 11 Line 17] and [Page 15 Line 2]
  • Many sentences are too long and hard to follow. Consider to simplify and cut many of them.
  • During the text, I am missing several pronouns (e.g. [Page 3 Line 9] “surface runoff sparse river network” should be “surface runoff, a sparse”; [Page 5 Line 22] Four lakes with “an” area increase). Misplaced prepositions (e.g. [Page 1 Line 37] “many of rivers and lakes”, the “of” is not necessary. Some words do not agree in number with other words of the phrase ([Page 1 Line 37] in “was the sources of” is “source). In general, English could use a review.

Round 2

Reviewer 1 Report

Dear Editor and authors,

After reading the revised version of the manuscript entitled: "The impact of permafrost degradation on lake changes in the endorheic basin on the Qinghai-Tibet Plateau", I have to report that the quality of the manuscript has significantly increased. The scientific story is more clear and the readability has been increased. However there are still some issues that I have with the manuscript that need minor correction, I have added these underneath:

  • In the version that I have the in-text references are missing. I could therefore not check the manuscript on scientific soundness.
  • The tense of verbs needs to be changed, some sentences are now in the past tense, while they should be in present/passive tense. Please let the manuscript be checked by a native English speaker.
  • On page 10 line 5 : "The simulated ALT was calibrated by measured data". I still do not know what this means. What kind of model did you use and how did you later come up with simulated values for ALT?
  • Please change the y-axis label of figure 7b and figure 8, so that they do not overlap with the numbers on the y-axis.
  • Correct the word "diagrammatic" in the caption of figure 9.
  • In the caption of Figure 10, please add that positive numbers are a decrease of ALT.

Discussion:

  • Are island-discontinuous the same as discontinuous-island? Both terms are used in the discussion. e.g. line 15 page 12.
  • Do not add pictures in the discussion. Either do it only in text (wording) or move the picture to the introduction, where you present the pathways to the reader.
  • What does "earth lake" on page 14, line 41 and 45 mean?
